# Functional connectivity changes in the brain of adolescents with internet addiction: A systematic literature review of imaging studies

Max L. Y. Chang[1], Irene O. Lee[2]*

1 Child and Adolescent Mental Health, Department of Brain Sciences, Great Ormond Street Institute of Child Health, University College London, London, United Kingdom, 2 Behavioural Brain Sciences Unit, Population Policy Practice Programme, Great Ormond Street Institute of Child Health, University College London, London, United Kingdom

* irene.lee@ucl.ac.uk

**Data Availability Statement:** All relevant data are within the paper and its Supporting information files.

## Abstract

Internet usage has seen a stark global rise over the last few decades, particularly among adolescents and young people, who have also been diagnosed increasingly with internet addiction (IA). IA impacts several neural networks that influence an adolescent's behaviour and development. This article issued a literature review on the resting-state and task-based functional magnetic resonance imaging (fMRI) studies to inspect the consequences of IA on the functional connectivity (FC) in the adolescent brain and its subsequent effects on their behaviour and development. A systematic search was conducted from two databases, PubMed and PsycINFO, to select eligible articles according to the inclusion and exclusion criteria. Eligibility criteria was especially stringent regarding the adolescent age range (10–19) and formal diagnosis of IA. Bias and quality of individual studies were evaluated. The fMRI results from 12 articles demonstrated that the effects of IA were seen throughout multiple neural networks: a mix of increases/decreases in FC in the default mode network; an overall decrease in FC in the executive control network; and no clear increase or decrease in FC within the salience network and reward pathway. The FC changes led to addictive behaviour and tendencies in adolescents. The subsequent behavioural changes are associated with the mechanisms relating to the areas of cognitive control, reward valuation, motor coordination, and the developing adolescent brain. Our results presented the FC alterations in numerous brain regions of adolescents with IA leading to the behavioural and developmental changes. Research on this topic had a low frequency with adolescent samples and were primarily produced in Asian countries. Future research studies of comparing results from Western adolescent samples provide more insight on therapeutic intervention.

## Introduction

The behavioural addiction brought on by excessive internet use has become a rising source of concern [1] since the last decade. According to clinical studies, individuals with Internet

**Funding:** The authors received no specific funding for this work.

**Competing interests:** The authors have declared that no competing interests exist.

Addiction (IA) or Internet Gaming Disorder (IGD) may have a range of biopsychosocial effects and is classified as an impulse-control disorder owing to its resemblance to pathological gambling and substance addiction [2, 3]. IA has been defined by researchers as a person's inability to resist the urge to use the internet, which has negative effects on their psychological well-being as well as their social, academic, and professional lives [4]. The symptoms can have serious physical and interpersonal repercussions and are linked to mood modification, salience, tolerance, impulsivity, and conflict [5]. In severe circumstances, people may experience severe pain in their bodies or health issues like carpal tunnel syndrome, dry eyes, irregular eating and disrupted sleep [6]. Additionally, IA is significantly linked to comorbidities with other psychiatric disorders [7].

Stevens et al (2021) reviewed 53 studies including 17 countries and reported the global prevalence of IA was 3.05% [8]. Asian countries had a higher prevalence (5.1%) than European countries (2.7%) [8]. Strikingly, adolescents and young adults had a global IGD prevalence rate of 9.9% which matches previous literature that reported historically higher prevalence among adolescent populations compared to adults [8, 9]. Over 80% of adolescent population in the UK, the USA, and Asia have direct access to the internet [10]. Children and adolescents frequently spend more time on media (possibly 7 hours and 22 minutes per day) than at school or sleeping [11]. Developing nations have also shown a sharp rise in teenage internet usage despite having lower internet penetration rates [10]. Concerns regarding the possible harms that overt internet use could do to adolescents and their development have arisen because of this surge, especially the significant impacts by the COVID-19 pandemic [12]. The growing prevalence and neurocognitive consequences of IA among adolescents makes this population a vital area of study [13].

Adolescence is a crucial developmental stage during which people go through significant changes in their biology, cognition, and personalities [14]. Adolescents' emotional-behavioural functioning is hyperactivated, which creates risk of psychopathological vulnerability [15]. In accordance with clinical study results [16], this emotional hyperactivity is supported by a high level of neuronal plasticity. This plasticity enables teenagers to adapt to the numerous physical and emotional changes that occur during puberty as well as develop communication techniques and gain independence [16]. However, the strong neuronal plasticity is also associated with risk-taking and sensation seeking [17] which may lead to IA.

Despite the fact that the precise neuronal mechanisms underlying IA are still largely unclear, functional magnetic resonance imaging (fMRI) method has been used by scientists as an important framework to examine the neuropathological changes occurring in IA, particularly in the form of functional connectivity (FC) [18]. fMRI research study has shown that IA alters both the functional and structural makeup of the brain [3].

We hypothesise that IA has widespread neurological alteration effects rather than being limited to a few specific brain regions. Further hypothesis holds that according to these alterations of FC between the brain regions or certain neural networks, adolescents with IA would experience behavioural changes. An investigation of these domains could be useful for creating better procedures and standards as well as minimising the negative effects of overt internet use. This literature review aims to summarise and analyse the evidence of various imaging studies that have investigated the effects of IA on the FC in adolescents. This will be addressed through two research questions:

1. How does internet addiction affect the functional connectivity in the adolescent brain?

2. How is adolescent behaviour and development impacted by functional connectivity changes due to internet addiction?

## Methods

The review protocol was conducted in line with the Preferred Reporting Items for Systematic Reviews and Meta-Analyses (PRISMA) guidelines (see S1 Checklist).

### Search strategy and selection process

A systematic search was conducted up until April 2023 from two sources of database, PubMed and PsycINFO, using a range of terms relevant to the title and research questions (see full list of search terms in S1 Appendix). All the searched articles can be accessed in the S1 Data. The eligible articles were selected according to the inclusion and exclusion criteria. Inclusion criteria used for the present review were: (i) participants in the studies with clinical diagnosis of IA; (ii) participants between the ages of 10 and 19; (iii) imaging research investigations; (iv) works published between January 2013 and April 2023; (v) written in English language; (vi) peer-reviewed papers and (vii) full text. The numbers of articles excluded due to not meeting the inclusion criteria are shown in Fig 1. Each study's title and abstract were screened for eligibility.

### Quality appraisal

Full texts of all potentially relevant studies were then retrieved and further appraised for eligibility. Furthermore, articles were critically appraised based on the GRADE (Grading of Recommendations, Assessment, Development, and Evaluations) framework to evaluate the individual study for both quality and bias. The subsequent quality levels were then appraised to each article and listed as either low, moderate, or high.

### Data collection process

Data that satisfied the inclusion requirements was entered into an excel sheet for data extraction and further selection. An article's author, publication year, country, age range, participant sample size, sex, area of interest, measures, outcome and article quality were all included in the data extraction spreadsheet. Studies looking at FC, for instance, were grouped, while studies looking at FC in specific area were further divided into sub-groups.

### Data synthesis and analysis

Articles were classified according to their location in the brain as well as the network or pathway they were a part of to create a coherent narrative between the selected studies. Conclusions concerning various research trends relevant to particular groupings were drawn from these groupings and subgroupings. To maintain the offered information in a prominent manner, these assertions were entered into the data extraction excel spreadsheet.

## Results

With the search performed on the selected databases, 238 articles in total were identified (see Fig 1). 15 duplicated articles were eliminated, and another 6 items were removed for various other reasons. Title and abstract screening eliminated 184 articles because they were not in English (number of article, n, = 7), did not include imaging components (n = 47), had adult participants (n = 53), did not have a clinical diagnosis of IA (n = 19), did not address FC in the brain (n = 20), and were published outside the desired timeframe (n = 38). A further 21 papers were eliminated for failing to meet inclusion requirements after the remaining 33 articles underwent full-text eligibility screening. A total of 12 papers were deemed eligible for this review analysis.

**Identification of studies via databases and registers**

**Identification**

Records identified from databases and manual searching: (n=238)
 PyscINFO (n = 111)
 PubMed (n = 127)

Records removed before screening:
- Duplicate records removed (n = 15)
- Records removed for other reasons (n = 6)

**Screening**

Records screened after duplicates were removed (n = 217)

Titles/Abstracts Screened (n = 217)

Records excluded: (n = 184)
- Not in English; 7
- Studies not including imaging component; 47
- Adult participant pool; 53
- No Internet Addiction clinical diagnosis; 19
- Not pertaining to functional connectivity in the brain; 20
- Out of timeframe: 38

**Eligibility**

Full text assessed for eligibility (n = 33)

21 full articles did not meet the inclusion criteria

**Included**

Studies included in review (n =12)

**Fig 1. Prisma flow diagram of study selection process.**

**Table 1. Data extraction sheet of the twelve selected articles.**

| Author(s) | Year | Country | Age Range | Sample Size | Sex | Area of Interest | Measures Used | Outcome | Quality Appraisal GRADE |
|---|---|---|---|---|---|---|---|---|---|
| **Hong et al.** [19] | 2013 | Korea | 11–16 | 12 | All Male | Whole brain | Resting State fMRI Imaging | Decreased FC in subcortical- frontal/ parietal areas for IA group | Moderate |
| **Lee et al.** [20] | 2020 | Korea | 12–15 | 17 | All Male | DMN, ECN, SN | Whole brain fMRI, seed to voxel analysis | Altered FC in DMN and SN in connection with left posterior superior temporal sulcus in IA group | Moderate-High |
| **Xing et al.** [21] | 2014 | China | 18–19 | 17 | 58.8% M, 41.1% F | SN | fMRI and colour-word Stroop task | Decreased FA in right SN in IA group, which may be related to poor behaviour performance in colour-word Stroop task | High |
| **Ding et al.** [22] | 2013 | China | 14–17 | 17 | 76% M, 24% F | PCC/ Whole Brain / DMN | Resting State fMRI Imaging | Increased FC in the bilateral cerebellum posterior lobe and middle temporal gyrus for IA group | Moderate-Low |
| **Hong et al.** [23] | 2015 | Korea | 11–16 | 12 | All Male | Putamen/ Caudate | Whole-brain echo-planar imaging | Altered dorsal putamen FC with the posterior insula-parietal operculum depending on IA diagnosis | Moderate-Low |
| **Wang et al.** [24] | 2017 | China | 14–16 | 31 | 68% M, 32% F | SN/interhemispheric/ DMN/ ECN (FPN) | Resting State fMRI Imaging | Reduced FC for DMN, SN, and interhemispheric interactions for IA adolescents | High |
| **Li et al.** [25] | 2014 | China | 13–17 | 23 | Not Given | Frontal Basal Ganglia | fMRI imaging and Go-Stop Task | Response inhibition within the frontal-basal ganglia pathway was not active in the IA group | Moderate-High |
| **Chen et al.** [26] | 2020 | China | 12–18 | 22 | 77% M, 23% F | Prefrontal striatal circuits | fMRI imaging and colour-word Stroop Task | Significant difference in FC in the between the left DLPFC and dorsal striatum for adolescents with IA | Moderate-High |
| **Wee et al.** [27] | 2014 | China | 15–19 | 17 | 88% M, 12%F | Whole-brain/ prefrontal/ occipital/ parietal | Resting State fMRI Imaging | Significant disruption of FC in the frontal, occipital, and parietal lobes for adolescents with IA | Moderate-High |
| **Li et al.** [28] | 2015 | China | 15–19 | 14 | 86% M, 14% F | Corticostriatal circuits | Resting State fMRI Imaging | Altered FC that correlated with neural functioning was found in IA subjects | Moderate |
| **Jin et al.** [29] | 2016 | China | 18–19 | 25 | 64% M, 36% F | Prefrontal cortex | Resting State fMRI Imaging | IA subjects showed decreased FC between several cortical regions and our seeds, including the insula, and temporal and occipital cortices. | Moderate |
| **Siste et al.** [30] | 2022 | Indonesia | 12–16 | 30 | 50% M, 50% F | General/ Prefrontal/ Insula | Resting State fMRI Imaging | FC of the LPFC(L) with the medial PFC, LPFC(L), as well as with the right lateral parietal cortex were lower for IA subjects | High |

Note: IA = Internet Addiction; FC = Functional Connectivity; fMRI = functional magnetic resonance imaging; DMN = Default Mode Network; ECN = Executive Control Network; SN = Salience Network; FPN = Fronto-Parietal Network; DLPFC = Dorsolateral Prefrontal Cortex; LPFC(L) = Lateral Prefrontal Cortex (Left).

Characteristics of the included studies, as depicted in the data extraction sheet in Table 1 provide information of the author(s), publication year, sample size, study location, age range, gender, area of interest, outcome, measures used and quality appraisal. Most of the studies in this review utilised resting state functional magnetic resonance imaging techniques (n = 7), with several studies demonstrating task-based fMRI procedures (n = 3), and the remaining studies utilising whole-brain imaging measures (n = 2). The studies were all conducted in Asiatic countries, specifically coming from China (8), Korea (3), and Indonesia (1). Sample sizes ranged from 12 to 31 participants with most of the imaging studies having comparable sample sizes. Majority of the studies included a mix of male and female participants (n = 8) with several studies having a male only participant pool (n = 3). All except one of the mixed gender studies had a majority male participant pool. One study did not disclose their data on the gender demographics of their experiment. Study years ranged from 2013–2022, with 2 studies in

2013, 3 studies in 2014, 3 studies in 2015, 1 study in 2017, 1 study in 2020, 1 study in 2021, and 1 study in 2022.

## (1) How does internet addiction affect the functional connectivity in the adolescent brain?

The included studies were organised according to the brain region or network that they were observing. The specific networks affected by IA were the default mode network, executive control system, salience network and reward pathway. These networks are vital components of adolescent behaviour and development [31]. The studies in each section were then grouped into subsections according to their specific brain regions within their network.

**Default mode network (DMN)/reward network.**   Out of the 12 studies, 3 have specifically studied the default mode network (DMN), and 3 observed whole-brain FC that partially included components of the DMN. The effect of IA on the various centres of the DMN was not unilaterally the same. The findings illustrate a complex mix of increases and decreases in FC depending on the specific region in the DMN (see Table 2 and Fig 2). The alteration of FC in posterior cingulate cortex (PCC) in the DMN was the most frequently reported area in adolescents with IA, which involved in attentional processes [32], but Lee et al. (2020) additionally found alterations of FC in other brain regions, such as anterior insula cortex, a node in the DMN that controls the integration of motivational and cognitive processes [20].

Ding et al. (2013) revealed altered FC in the cerebellum, the middle temporal gyrus, and the medial prefrontal cortex (mPFC) [22]. They found that the bilateral inferior parietal lobule, left superior parietal lobule, and right inferior temporal gyrus had decreased FC, while the bilateral posterior lobe of the cerebellum and the medial temporal gyrus had increased FC [22]. The right middle temporal gyrus was found to have 111 cluster voxels (t = 3.52, p<0.05) and the right inferior parietal lobule was found to have 324 cluster voxels (t = -4.07, p<0.05) with an extent threshold of 54 voxels (figures above this threshold are deemed significant) [22]. Additionally, there was a negative correlation, with 95 cluster voxels (p<0.05) between the FC of the left superior parietal lobule and the PCC with the Chen Internet Addiction Scores (CIAS) which are used to determine the severity of IA [22]. On the other hand, in regions of the reward system, connection with the PCC was positively connected with CIAS scores [22]. The most significant was the right praecuneus with 219 cluster voxels (p<0.05) [22]. Wang et al. (2017) also discovered that adolescents with IA had 33% less FC in the left inferior parietal lobule and 20% less FC in the dorsal mPFC [24]. A potential connection between the effects of substance use and overt internet use is revealed by the generally decreased FC in these areas of the DMN of teenagers with drug addiction and IA [35].

The putamen was one of the main regions of reduced FC in adolescents with IA [19]. The putamen and the insula-operculum demonstrated significant group differences regarding functional connectivity with a cluster size of 251 and an extent threshold of 250 (Z = 3.40, p<0.05) [19]. The molecular mechanisms behind addiction disorders have been intimately connected to decreased striatal dopaminergic function [19], making this function crucial.

**Executive Control Network (ECN).**   5 studies out of 12 have specifically viewed parts of the executive control network (ECN) and 3 studies observed whole-brain FC. The effects of IA on the ECN's constituent parts were consistent across all the studies examined for this analysis (see Table 2 and Fig 3). The results showed a notable decline in all the ECN's major centres. Li et al. (2014) used fMRI imaging and a behavioural task to study response inhibition in adolescents with IA [25] and found decreased activation at the striatum and frontal gyrus, particularly a reduction in FC at inferior frontal gyrus, in the IA group compared to controls [25]. The inferior frontal gyrus showed a reduction in FC in comparison to the controls with a

**Table 2. The effects of internet addiction on the functional connectivity in adolescent brain networks/regions affecting their functions.** The overall changes of functional connectivity in the brain network including default mode network (DMN), executive control network (ECN), salience network (SN) and reward network. IA = Internet Addiction, FC = Functional Connectivity.

| Brain Network/Region | Effects of IA on FC | Function | References |
|---|---|---|---|
| **Default Mode Network (DMN)/ Rewards Network** | | | |
| Posterior cingulate cortex (PCC) | Altered | Attention processes | [20, 32] |
| | Negative correlation between FC and severity of IA | | [22] |
| Right inferior temporal gyrus | Decreased | Semantic memory processing, visual perception | [22] |
| Medial temporal gyrus | Increased | Visual-motion component | [22] |
| Medial prefrontal cortex | Altered | Aids in context formation | [22] |
| Dorsal Medial prefrontal cortex | Decreased | Cognitive functioning, motor control, reward valuation | [24] |
| Parietal lobule (Bilateral inferior & left superior) | Decreased | Integrates sensory information | [22, 24] |
| Parietal lobule (left superior) | Negative correlation between FC and severity of IA | Attention and visuospatial perception | [22] |
| Bilateral posterior lobe of the cerebellum | Increased | Motor coordination and inhibition | [22] |
| Left inferior parietal lobule | Decreased | Semantic information processing | [24] |
| Putamen | Decreased | Motor control and learning | [19] |
| Regions of Reward system (connect with PCC): Include right praecuneus, posterior cingulate gyrus, thalamus, caudate, nucleus accumbens, supplementary motor area, and lingual gyrus | Positive correlation between FC and severity of IA | Reward amount, receipt, and direction (Visual-spatial) | [22] |
| Anterior Insula | Altered | Motivational and cognitive processes | [20] |
| Insula | Decreased | Salience processing, Interoception, Craving | [24] |
| **Executive Control Network** | | | |
| Striatum | Decreased | Attention | [25, 33] |
| Dorsal Striatum | Decreased | Attention, Reward, Decision-making, Motivation | [26, 29] |
| Frontal gyrus | Decreased | Reward, Decision-making, Motivation | [25] |
| Frontal-basal ganglia | Decreased | Cognition, adaptive function, and motor coordination | [25] |
| Left dorsolateral prefrontal cortex | Decreased | Attention switching, working memory, planning | [26, 29] |
| Caudate | Decreased | Movement, learning, motivation, reward valuation | [28, 34] |
| Bilateral anterior cingulate cortex | Decreased | Decision-making, attention, impulse-control, emotional processing | [28, 34] |
| Insula | Decreased | Salience processing, Interoception, Craving | [29] |
| Temporal cortices | Decreased | Encoding memory, processing sensory information | [29] |
| Thalamus | Decreased | Sensory relay, consciousness | [29] |
| **Salience Network (SN)** | Decreased structural connectivity and fractional anisotropy but not FC | Altered cognitive control, salience to stimuli, conflict monitoring, interoceptive states of consciousness | [21] |
| Connection between SN and anterior DMN | Decreased | Performance of socio-cognitive tasks | [24] |

cluster size of 71 (t = 4.18, p<0.05) [25]. In addition, the frontal-basal ganglia pathways in the adolescents with IA showed little effective connection between areas and increased degrees of response inhibition [25].

Lin et al. (2015) found that adolescents with IA demonstrated disrupted corticostriatal FC compared to controls [33]. The corticostriatal circuitry experienced decreased connectivity with the caudate, bilateral anterior cingulate cortex (ACC), as well as the striatum and frontal

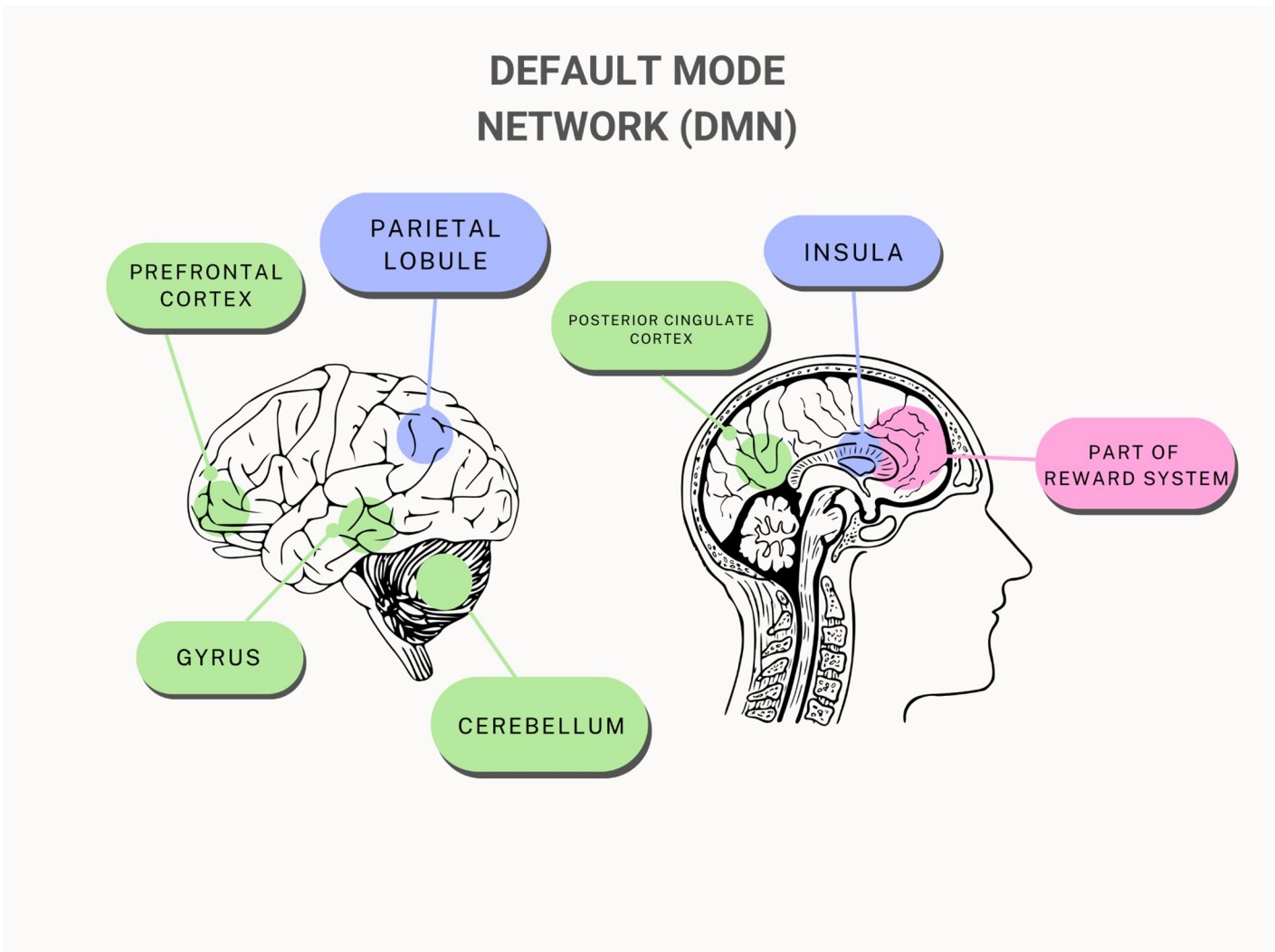

**Fig 2. General overview of the brain regions in the default mode network affected by internet addiction.**

gyrus [33]. The inferior ventral striatum showed significantly reduced FC with the subcallosal ACC and caudate head with cluster size of 101 (t = -4.64, p<0.05) [33]. Decreased FC in the caudate implies dysfunction of the corticostriatal-limbic circuitry involved in cognitive and emotional control [36]. The decrease in FC in both the striatum and frontal gyrus is related to inhibitory control, a common deficit seen with disruptions with the ECN [33].

The dorsolateral prefrontal cortex (DLPFC), ACC, and right supplementary motor area (SMA) of the prefrontal cortex were all found to have significantly decreased grey matter volume [29]. In addition, the DLPFC, insula, temporal cortices, as well as significant subcortical regions like the striatum and thalamus, showed decreased FC [29]. According to Tremblay (2009), the striatum plays a significant role in the processing of rewards, decision-making, and motivation [37]. Chen et al. (2020) reported that the IA group demonstrated increased impulsivity as well as decreased reaction inhibition using a Stroop colour-word task [26]. Furthermore, Chen et al. (2020) observed that the left DLPFC and dorsal striatum experienced a negative connection efficiency value, specifically demonstrating that the dorsal striatum activity suppressed the left DLPFC [27].

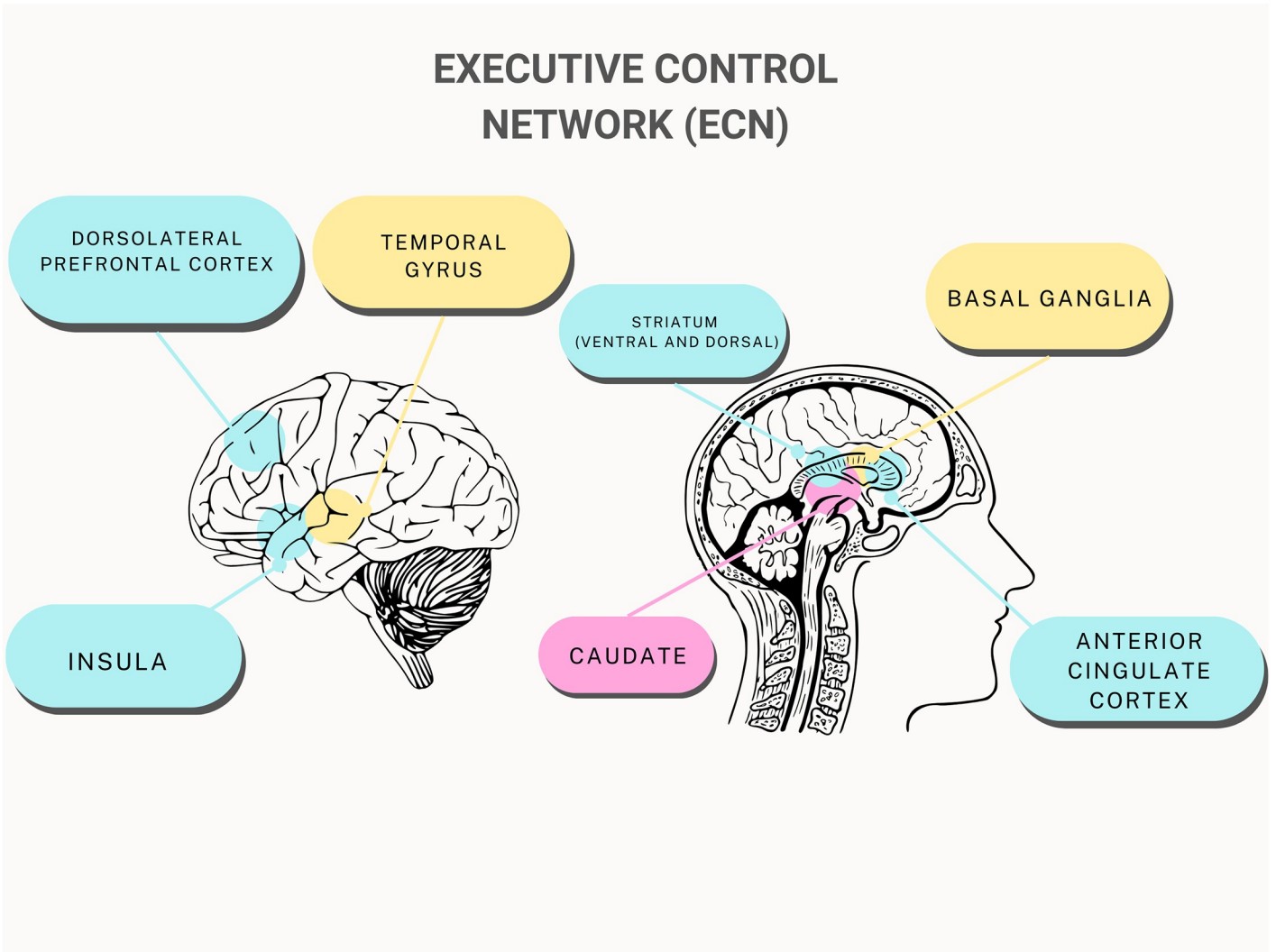

**Fig 3. General overview of the brain regions in the executive control network affected by internet addiction.**

**Salience network (SN).** Out of the 12 chosen studies, 3 studies specifically looked at the salience network (SN) and 3 studies have observed whole-brain FC. Relative to the DMN and ECN, the findings on the SN were slightly sparser. Despite this, adolescents with IA demonstrated a moderate decrease in FC, as well as other measures like fibre connectivity and cognitive control, when compared to healthy control (see Table 2 and Fig 4).

Xing et al. (2014) used both dorsal anterior cingulate cortex (dACC) and insula to test FC changes in the SN of adolescents with IA and found decreased structural connectivity in the SN as well as decreased fractional anisotropy (FA) that correlated to behaviour performance in the Stroop colour word-task [21]. They examined the dACC and insula to determine whether the SN's disrupted connectivity may be linked to the SN's disruption of regulation, which would explain the impaired cognitive control seen in adolescents with IA. However, researchers did not find significant FC differences in the SN when compared to the controls [21]. These results provided evidence for the structural changes in the interconnectivity within SN in adolescents with IA.

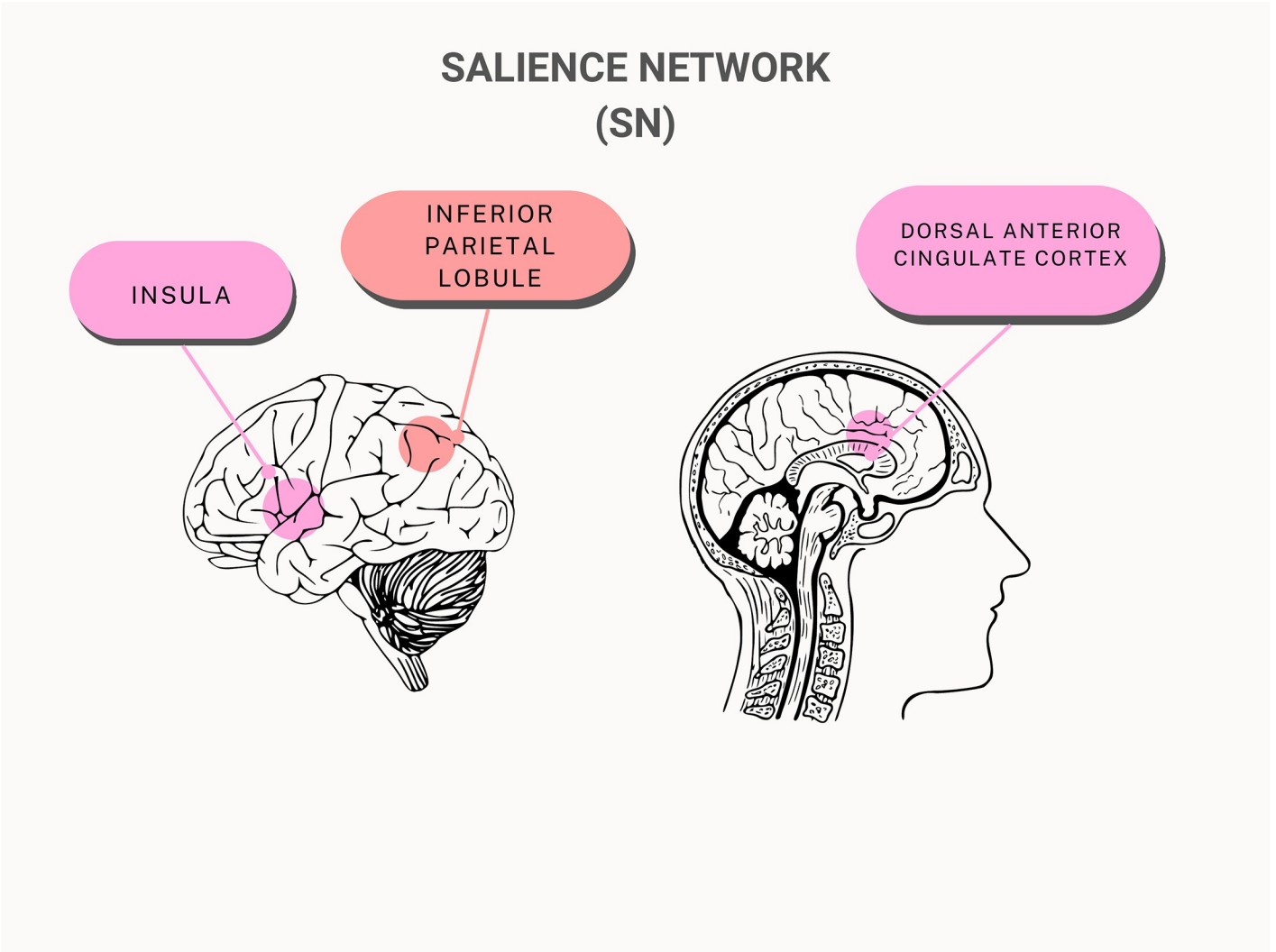

**Fig 4. General overview of the brain regions in the salience network affected by internet addiction.**

Wang et al. (2017) investigated network interactions between the DMN, ECN, SN and reward pathway in IA subjects [24] (see Fig 5), and found 40% reduction of FC between the DMN and specific regions of the SN, such as the insula, in comparison to the controls (p = 0.008) [24]. The anterior insula and dACC are two areas that are impacted by this altered FC [24]. This finding supports the idea that IA has similar neurobiological abnormalities with other addictive illnesses, which is in line with a study that discovered disruptive changes in the SN and DMN's interaction in cocaine addiction [38]. The insula has also been linked to the intensity of symptoms and has been implicated in the development of IA [39].

### (2) How is adolescent behaviour and development impacted by functional connectivity changes due to internet addiction?

**Default mode network (DMN)/reward network.** The findings that IA individuals demonstrate an overall decrease in FC in the DMN is supported by numerous research [24]. Drug addict populations also exhibited similar decline in FC in the DMN [40]. The disruption of

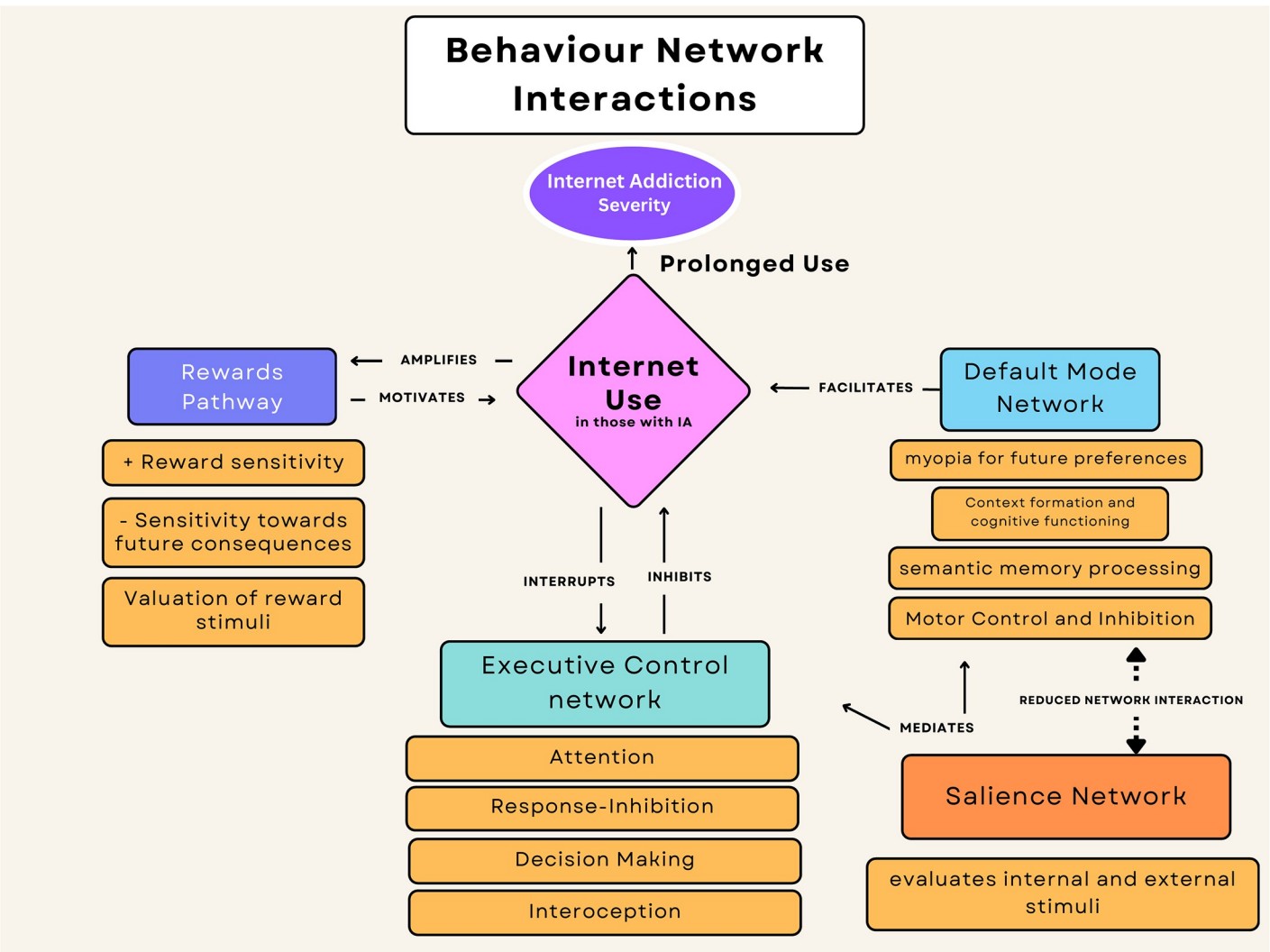

**Fig 5. Behaviour network interactions diagram depicts the behaviour network interactions between the default mode network (DMN), executive control network (ECN), salience network (SN) and reward pathway during the use of internet in those with IA.** "+" indicates an increase in behaivour; "-"indicates a decrease in behaviour; solid arrows indicate a direct network interaction; and the dotted arrows indicates a reduction in network interaction. This diagram depicts network interactions juxtaposed with engaging in internet related behaviours. Through the neural interactions, the diagram illustrates how the networks inhibit or amplify internet usage and vice versa. Furthermore, it demonstrates how the SN mediates both the DMN and ECN.

attentional orientation and self-referential processing for both substance and behavioural addiction was then hypothesised to be caused by DMN anomalies in FC [41].

In adolescents with IA, decline of FC in the parietal lobule affects visuospatial task-related behaviour [22], short-term memory [42], and the ability of controlling attention or restraining motor responses during response inhibition tests [42]. Cue-induced gaming cravings are influenced by the DMN [43]. A visual processing area called the praecuneus links gaming cues to internal information [22]. A meta-analysis found that the posterior cingulate cortex activity of individuals with IA during cue-reactivity tasks was connected with their gaming time [44], suggesting that excessive gaming may impair DMN function and that individuals with IA exert more cognitive effort to control it. Findings for the behavioural consequences of FC changes in the DMN illustrate its underlying role in regulating impulsivity, self-monitoring, and cognitive control.

Furthermore, Ding et al. (2013) reported an activation of components of the reward pathway, including areas like the nucleus accumbens, praecuneus, SMA, caudate, and thalamus, in connection to the DMN [22]. The increased FC of the limbic and reward networks have been confirmed to be a major biomarker for IA [45, 46]. The increased reinforcement in these networks increases the strength of reward stimuli and makes it more difficult for other networks, namely the ECN, to down-regulate the increased attention [29] (See Fig 5).

**Executive control network (ECN).** The numerous IA-affected components in the ECN have a role in a variety of behaviours that are connected to both response inhibition and emotional regulation [47]. For instance, brain regions like the striatum, which are linked to impulsivity and the reward system, are heavily involved in the act of playing online games [47]. Online game play activates the striatum, which suppresses the left DLPFC in ECN [48]. As a result, people with IA may find it difficult to control their want to play online games [48]. This system thus causes impulsive and protracted gaming conduct, lack of inhibitory control leading to the continued use of internet in an overt manner despite a variety of negative effects, personal distress, and signs of psychological dependence [33] (See Fig 5).

Wang et al. (2017) report that disruptions in cognitive control networks within the ECN are frequently linked to characteristics of substance addiction [24]. With samples that were addicted to heroin and cocaine, previous studies discovered abnormal FC in the ECN and the PFC [49]. Electronic gaming is known to promote striatal dopamine release, similar to drug addiction [50]. According to Drgonova and Walther (2016), it is hypothesised that dopamine could stimulate the reward system of the striatum in the brain, leading to a loss of impulse control and a failure of prefrontal lobe executive inhibitory control [51]. In the end, IA's resemblance to drug use disorders may point to vital biomarkers or underlying mechanisms that explain how cognitive control and impulsive behaviour are related.

A task-related fMRI study found that the decrease in FC between the left DLPFC and dorsal striatum was congruent with an increase in impulsivity in adolescents with IA [26]. The lack of response inhibition from the ECN results in a loss of control over internet usage and a reduced capacity to display goal-directed behaviour [33]. Previous studies have linked the alteration of the ECN in IA with higher cue reactivity and impaired ability to self-regulate internet specific stimuli [52].

**Salience network (SN)/ other networks.** Xing et al. (2014) investigated the significance of the SN regarding cognitive control in teenagers with IA [21]. The SN, which is composed of the ACC and insula, has been demonstrated to control dynamic changes in other networks to modify cognitive performance [21]. The ACC is engaged in conflict monitoring and cognitive control, according to previous neuroimaging research [53]. The insula is a region that integrates interoceptive states into conscious feelings [54]. The results from Xing et al. (2014) showed declines in the SN regarding its structural connectivity and fractional anisotropy, even though they did not observe any appreciable change in FC in the IA participants [21]. Due to the small sample size, the results may have indicated that FC methods are not sensitive enough to detect the significant functional changes [21]. However, task performance behaviours associated with impaired cognitive control in adolescents with IA were correlated with these findings [21]. Our comprehension of the SN's broader function in IA can be enhanced by this relationship.

Research study supports the idea that different psychological issues are caused by the functional reorganisation of expansive brain networks, such that strong association between SN and DMN may provide neurological underpinnings at the system level for the uncontrollable character of internet-using behaviours [24]. In the study by Wang et al. (2017), the decreased interconnectivity between the SN and DMN, comprising regions such the DLPFC and the insula, suggests that adolescents with IA may struggle to effectively inhibit DMN activity

during internally focused processing, leading to poorly managed desires or preoccupations to use the internet [24] (See Fig 5). Subsequently, this may cause a failure to inhibit DMN activity as well as a restriction of ECN functionality [55]. As a result, the adolescent experiences an increased salience and sensitivity towards internet addicting cues making it difficult to avoid these triggers [56].

## Discussion

The primary aim of this review was to present a summary of how internet addiction impacts on the functional connectivity of adolescent brain. Subsequently, the influence of IA on the adolescent brain was compartmentalised into three sections: alterations of FC at various brain regions, specific FC relationships, and behavioural/developmental changes. Overall, the specific effects of IA on the adolescent brain were not completely clear, given the variety of FC changes. However, there were overarching behavioural, network and developmental trends that were supported that provided insight on adolescent development.

The first hypothesis that was held about this question was that IA was widespread and would be regionally similar to substance-use and gambling addiction. After conducting a review of the information in the chosen articles, the hypothesis was predictably supported. The regions of the brain affected by IA are widespread and influence multiple networks, mainly DMN, ECN, SN and reward pathway. In the DMN, there was a complex mix of increases and decreases within the network. However, in the ECN, the alterations of FC were more unilaterally decreased, but the findings of SN and reward pathway were not quite clear. Overall, the FC changes within adolescents with IA are very much network specific and lay a solid foundation from which to understand the subsequent behaviour changes that arise from the disorder.

The second hypothesis placed emphasis on the importance of between network interactions and within network interactions in the continuation of IA and the development of its behavioural symptoms. The results from the findings involving the networks, DMN, SN, ECN and reward system, support this hypothesis (see Fig 5). Studies confirm the influence of all these neural networks on reward valuation, impulsivity, salience to stimuli, cue reactivity and other changes that alter behaviour towards the internet use. Many of these changes are connected to the inherent nature of the adolescent brain.

There are multiple explanations that underlie the vulnerability of the adolescent brain towards IA related urges. Several of them have to do with the inherent nature and underlying mechanisms of the adolescent brain. Children's emotional, social, and cognitive capacities grow exponentially during childhood and adolescence [57]. Early teenagers go through a process called "social reorientation" that is characterised by heightened sensitivity to social cues and peer connections [58]. Adolescents' improvements in their social skills coincide with changes in their brains' anatomical and functional organisation [59]. Functional hubs exhibit growing connectivity strength [60], suggesting increased functional integration during development. During this time, the brain's functional networks change from an anatomically dominant structure to a scattered architecture [60].

The adolescent brain is very responsive to synaptic reorganisation and experience cues [61]. As a result, one of the distinguishing traits of the maturation of adolescent brains is the variation in neural network trajectory [62]. Important weaknesses of the adolescent brain that may explain the neurobiological change brought on by external stimuli are illustrated by features like the functional gaps between networks and the inadequate segregation of networks [62].

The implications of these findings towards adolescent behaviour are significant. Although the exact changes and mechanisms are not fully clear, the observed changes in functional connectivity have the capacity of influencing several aspects of adolescent development. For

example, functional connectivity has been utilised to investigate attachment styles in adolescents [63]. It was observed that adolescent attachment styles were negatively associated with caudate-prefrontal connectivity, but positively with the putamen-visual area connectivity [63]. Both named areas were also influenced by the onset of internet addiction, possibly providing a connection between the two. Another study associated neighbourhood/socioeconomic disadvantage with functional connectivity alterations in the DMN and dorsal attention network [64]. The study also found multivariate brain behaviour relationships between the altered/disadvantaged functional connectivity and mental health and cognition [64]. This conclusion supports the notion that the functional connectivity alterations observed in IA are associated with specific adolescent behaviours as well as the fact that functional connectivity can be utilised as a platform onto which to compare various neurologic conditions.

## Limitations/strengths

There were several limitations that were related to the conduction of the review as well as the data extracted from the articles. Firstly, the study followed a systematic literature review design when analysing the fMRI studies. The data pulled from these imaging studies were namely qualitative and were subject to bias contrasting the quantitative nature of statistical analysis. Components of the study, such as sample sizes, effect sizes, and demographics were not weighted or controlled. The second limitation brought up by a similar review was the lack of a universal consensus of terminology given IA [47]. Globally, authors writing about this topic use an array of terminology including online gaming addiction, internet addiction, internet gaming disorder, and problematic internet use. Often, authors use multiple terms interchangeably which makes it difficult to depict the subtle similarities and differences between the terms.

Reviewing the explicit limitations in each of the included studies, two major limitations were brought up in many of the articles. One was relating to the cross-sectional nature of the included studies. Due to the inherent qualities of a cross-sectional study, the studies did not provide clear evidence that IA played a causal role towards the development of the adolescent brain. While several biopsychosocial factors mediate these interactions, task-based measures that combine executive functions with imaging results reinforce the assumed connection between the two that is utilised by the papers studying IA. Another limitation regarded the small sample size of the included studies, which averaged to around 20 participants. The small sample size can influence the generalisation of the results as well as the effectiveness of statistical analyses. Ultimately, both included study specific limitations illustrate the need for future studies to clarify the causal relationship between the alterations of FC and the development of IA.

Another vital limitation was the limited number of studies applying imaging techniques for investigations on IA in adolescents were a uniformly Far East collection of studies. The reason for this was because the studies included in this review were the only fMRI studies that were found that adhered to the strict adolescent age restriction. The adolescent age range given by the WHO (10–19 years old) [65] was strictly followed. It is important to note that a multitude of studies found in the initial search utilised an older adolescent demographic that was slightly higher than the WHO age range and had a mean age that was outside of the limitations. As a result, the results of this review are biased and based on the 12 studies that met the inclusion and exclusion criteria.

Regarding the global nature of the research, although the journals that the studies were published in were all established western journals, the collection of studies were found to all originate from Asian countries, namely China and Korea. Subsequently, it pulls into question if the results and measures from these studies are generalisable towards a western population. As

stated previously, Asian countries have a higher prevalence of IA, which may be the reasoning to why the majority of studies are from there [8]. However, in an additional search including other age groups, it was found that a high majority of all FC studies on IA were done in Asian countries. Interestingly, western papers studying fMRI FC were primarily focused on gambling and substance-use addiction disorders. The western papers on IA were less focused on fMRI FC but more on other components of IA such as sleep, game-genre, and other non-imaging related factors. This demonstrated an overall lack of western fMRI studies on IA. It is important to note that both western and eastern fMRI studies on IA presented an overall lack on children and adolescents in general.

Despite the several limitations, this review provided a clear reflection on the state of the data. The strengths of the review include the strict inclusion/exclusion criteria that filtered through studies and only included ones that contained a purely adolescent sample. As a result, the information presented in this review was specific to the review's aims. Given the sparse nature of adolescent specific fMRI studies on the FC changes in IA, this review successfully provided a much-needed niche representation of adolescent specific results. Furthermore, the review provided a thorough functional explanation of the DMN, ECN, SN and reward pathway making it accessible to readers new to the topic.

## Future directions and implications

Through the search process of the review, there were more imaging studies focused on older adolescence and adulthood. Furthermore, finding a review that covered a strictly adolescent population, focused on FC changes, and was specifically depicting IA, was proven difficult. Many related reviews, such as Tereshchenko and Kasparov (2019), looked at risk factors related to the biopsychosocial model, but did not tackle specific alterations in specific structural or functional changes in the brain [66]. Weinstein (2017) found similar structural and functional results as well as the role IA has in altering response inhibition and reward valuation in adolescents with IA [47]. Overall, the accumulated findings only paint an emerging pattern which aligns with similar substance-use and gambling disorders. Future studies require more specificity in depicting the interactions between neural networks, as well as more literature on adolescent and comorbid populations. One future field of interest is the incorporation of more task-based fMRI data. Advances in resting-state fMRI methods have yet to be reflected or confirmed in task-based fMRI methods [62]. Due to the fact that network connectivity is shaped by different tasks, it is critical to confirm that the findings of the resting state fMRI studies also apply to the task based ones [62]. Subsequently, work in this area will confirm if intrinsic connectivity networks function in resting state will function similarly during goal directed behaviour [62]. An elevated focus on adolescent populations as well as task-based fMRI methodology will help uncover to what extent adolescent network connectivity maturation facilitates behavioural and cognitive development [62].

A treatment implication is the potential usage of bupropion for the treatment of IA. Bupropion has been previously used to treat patients with gambling disorder and has been effective in decreasing overall gambling behaviour as well as money spent while gambling [67]. Bae et al. (2018) found a decrease in clinical symptoms of IA in line with a 12-week bupropion treatment [31]. The study found that bupropion altered the FC of both the DMN and ECN which in turn decreased impulsivity and attentional deficits for the individuals with IA [31]. Interventions like bupropion illustrate the importance of understanding the fundamental mechanisms that underlie disorders like IA.

## Conclusion

The goal for this review was to summarise the current literature on functional connectivity changes in adolescents with internet addiction. The findings answered the primary research questions that were directed at FC alterations within several networks of the adolescent brain and how that influenced their behaviour and development. Overall, the research demonstrated several wide-ranging effects that influenced the DMN, SN, ECN, and reward centres. Additionally, the findings gave ground to important details such as the maturation of the adolescent brain, the high prevalence of Asian originated studies, and the importance of task-based studies in this field. The process of making this review allowed for a thorough understanding IA and adolescent brain interactions.

Given the influx of technology and media in the lives and education of children and adolescents, an increase in prevalence and focus on internet related behavioural changes is imperative towards future children/adolescent mental health. Events such as COVID-19 act to expose the consequences of extended internet usage on the development and lifestyle of specifically young people. While it is important for parents and older generations to be wary of these changes, it is important for them to develop a base understanding of the issue and not dismiss it as an all-bad or all-good scenario. Future research on IA will aim to better understand the causal relationship between IA and psychological symptoms that coincide with it. The current literature regarding functional connectivity changes in adolescents is limited and requires future studies to test with larger sample sizes, comorbid populations, and populations outside Far East Asia.

This review aimed to demonstrate the inner workings of how IA alters the connection between the primary behavioural networks in the adolescent brain. Predictably, the present answers merely paint an unfinished picture that does not necessarily depict internet usage as overwhelmingly positive or negative. Alternatively, the research points towards emerging patterns that can direct individuals on the consequences of certain variables or risk factors. A clearer depiction of the mechanisms of IA would allow physicians to screen and treat the onset of IA more effectively. Clinically, this could be in the form of more streamlined and accurate sessions of CBT or family therapy, targeting key symptoms of IA. Alternatively clinicians could potentially prescribe treatment such as bupropion to target FC in certain regions of the brain. Furthermore, parental education on IA is another possible avenue of prevention from a public health standpoint. Parents who are aware of the early signs and onset of IA will more effectively handle screen time, impulsivity, and minimize the risk factors surrounding IA.

Additionally, an increased attention towards internet related fMRI research is needed in the West, as mentioned previously. Despite cultural differences, Western countries may hold similarities to the eastern countries with a high prevalence of IA, like China and Korea, regarding the implications of the internet and IA. The increasing influence of the internet on the world may contribute to an overall increase in the global prevalence of IA. Nonetheless, the high saturation of eastern studies in this field should be replicated with a Western sample to determine if the same FC alterations occur. A growing interest in internet related research and education within the West will hopefully lead to the knowledge of healthier internet habits and coping strategies among parents with children and adolescents. Furthermore, IA research has the potential to become a crucial proxy for which to study adolescent brain maturation and development.

## Supporting information

**S1 Checklist. PRISMA checklist.**
(DOCX)

**S1 Appendix. Search strategies with all the terms.**
(DOCX)

**S1 Data. Article screening records with details of categorized content.**
(XLSX)

## Acknowledgments

The authors thank https://www.stockio.com/free-clipart/brain-01 (with attribution to Stockio.com); and https://www.rawpixel.com/image/6442258/png-sticker-vintage for the free images used to create Figs 2–4.

## Author Contributions

**Conceptualization:** Max L. Y. Chang, Irene O. Lee.

**Data curation:** Max L. Y. Chang.

**Formal analysis:** Max L. Y. Chang.

**Investigation:** Max L. Y. Chang.

**Methodology:** Max L. Y. Chang.

**Project administration:** Max L. Y. Chang.

**Software:** Max L. Y. Chang.

**Supervision:** Irene O. Lee.

**Validation:** Max L. Y. Chang, Irene O. Lee.

**Visualization:** Max L. Y. Chang.

**Writing – original draft:** Max L. Y. Chang.

**Writing – review & editing:** Max L. Y. Chang, Irene O. Lee.

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
