## [Decision Letter · Decision Letter 0]

6 Feb 2024

PMEN-D-23-00073

Functional connectivity changes in the brain of adolescents with internet addiction: A systematic literature review of imaging studies

PLOS Mental Health

Dear Dr. Lee,

Thank you for submitting your manuscript to PLOS Mental Health. After careful consideration, we feel that it has merit but does not fully meet PLOS Mental Health’s publication criteria as it currently stands. Therefore, we invite you to submit a revised version of the manuscript that addresses the points raised during the review process.

Please submit your revised manuscript by **March 5th, 2024.** If you will need more time than this to complete your revisions, please reply to this message or contact the journal office at mentalhealth@plos.org. Please include the following items when submitting your revised manuscript:

We look forward to receiving your revised manuscript.

Kind regards,

Dr Omona Kizito, PhD

Academic Editor

PLOS Mental Health

Journal Requirements:

1. Please note that your Data Availability Statement is currently missing [the repository name and/or the DOI/accession number of each dataset OR a direct link to access each database]. If your manuscript is accepted for publication, you will be asked to provide these details on a very short timeline. We therefore suggest that you provide this information now, though we will not hold up the peer review process if you are unable.

2. We have noticed that you have uploaded Supporting Information files, but you have not included a list of legends. Please add a full list of legends for your Supporting Information files after the references list. 

Additional Editor Comments (if provided):

Reviewers' comments:

Reviewer's Responses to Questions

**Comments to the Author**

1. Does this manuscript meet PLOS Mental Health’s publication criteria? Is the manuscript technically sound, and do the data support the conclusions? The manuscript must describe methodologically and ethically rigorous research with conclusions that are appropriately drawn based on the data presented.

Reviewer #1: Yes

Reviewer #2: Yes

Reviewer #3: Yes

2. Has the statistical analysis been performed appropriately and rigorously?

Reviewer #1: No

Reviewer #2: N/A

Reviewer #3: N/A

3. Have the authors made all data underlying the findings in their manuscript fully available (please refer to the Data Availability Statement at the start of the manuscript PDF file)?

Reviewer #1: No

Reviewer #2: Yes

Reviewer #3: No

4. Is the manuscript presented in an intelligible fashion and written in standard English?

Reviewer #1: Yes

Reviewer #2: Yes

Reviewer #3: Yes

5. Review Comments to the Author

Reviewer #1: The manuscript addresses a contemporary and significant issue, namely internet addiction among adolescents. The rising global internet usage and its impact on neural networks make this study relevant and timely. The systematic search from reputable databases (PubMed and PsycINFO) and the strict adherence to inclusion/exclusion criteria, particularly regarding the adolescent age range and formal diagnosis of internet addiction, contribute to the methodological rigor of the review. The literature review covers both resting-state and task-based functional magnetic resonance imaging (fMRI) studies, providing a comprehensive overview of the consequences of internet addiction on the functional connectivity in the adolescent brain. The fMRI results are succinctly presented, detailing the effects of internet addiction on various neural networks. The delineation of increases/decreases in functional connectivity within specific networks adds clarity to the findings. The manuscript effectively connects changes in functional connectivity to subsequent behavioral alterations in adolescents, emphasizing the relevance of the neural changes to addictive behavior and tendencies. However there are certain areas that need to be improved in the manuscript are listed below:

1. The discussion could be enhanced by delving deeper into the implications of the findings. Elaborating on how the observed changes in functional connectivity may influence specific behaviors and developmental aspects would provide a more nuanced understanding.

2. It would be beneficial to explicitly acknowledge any limitations of the reviewed studies and the broader implications for the interpretation of the results. Discussing potential biases or methodological constraints would strengthen the manuscript.

3. Including a section on potential future research directions or interventions based on the findings would add value to the manuscript. This could stimulate further discourse on addressing internet addiction in adolescents.

4. The conclusion could be strengthened by summarizing the key insights and emphasizing the broader implications for future research, clinical applications, or public health interventions.

Reviewer #2: The manuscript was well written and presented in an intelligible manner. Very interesting topic and I believed that the article will contribute more to the field of internet addiction vs adolescent mental health. I think the manuscript should be published without any revision.

Reviewer #3: Thank you for submitting your work to Plos Mental health journal.

The research challenge is not adequately addressed in the introduction section. rather than providing a basic summary, researchers would be better served by doing a meta-analysis to address any issues that were either overlooked or not addressed in the original studies. A comprehensive meta-analysis would allow researchers to examine the functional connectivity changes in the brains of adolescents with internet addiction, while also providing an in-depth analysis of the pros and cons of related issues. This approach would provide a more thorough understanding of the topic and help to identify any gaps or limitations in the existing literature. By conducting a comprehensive meta-analysis, researchers can not only delve into the functional connectivity changes in the brains of adolescents with internet addiction but also explore a wide range of related topics, such as the impact of internet addiction on cognitive function, mental health, and social interactions. This approach would offer a more comprehensive understanding of the subject matter and enable the identification of any gaps or limitations in the existing literature, thus contributing to the advancement of research in this field.

The study analyzed functional connectivity changes in the brains of adolescents with internet addiction, but no thematic analysis was presented. Without presenting any thematic analysis, it was difficult to draw conclusions from the study. An in-depth thematic analysis would have allowed the researchers to identify patterns and trends in the functional connectivity changes, which would have allowed them to better understand the causes and effects of internet addiction. Further research is needed to explore the specific neural networks affected by internet addiction and their implications for cognitive and emotional functioning. This is because the thematic analysis would have allowed the researchers to extract the key themes and patterns from the data, which could then be interpreted and used to draw meaningful conclusions. Additionally, an in-depth analysis of the neural networks affected by internet addiction could provide a better understanding of how internet addiction affects cognitive and emotional functioning. Furthermore, global reviews and meta-analyses could be included to derive distinct viewpoints and consolidate findings across multiple studies. This would enhance the robustness of the conclusions and provide a more comprehensive understanding of the functional connectivity changes associated with internet addiction in adolescents.

6. PLOS authors have the option to publish the peer review history of their article (what does this mean?). If published, this will include your full peer review and any attached files.

**Do you want your identity to be public for this peer review?** For information about this choice, including consent withdrawal, please see our Privacy Policy.

Reviewer #1: No

Reviewer #2: **Yes: **Victor Adeleke

Reviewer #3: No

---

## [Editor Report · Decision Letter 1]

26 Mar 2024

Functional connectivity changes in the brain of adolescents with internet addiction: A systematic literature review of imaging studies

PMEN-D-23-00073R1

Dear  Ms Lee,

We are pleased to inform you that your manuscript titled 'Functional connectivity changes in the brain of adolescents with internet addiction: A systematic literature review of imaging studies' has been provisionally accepted for publication in PLOS Mental Health.

Before your manuscript can be formally accepted, you will need to complete some formatting changes, which you will receive in a follow up email. A member of our team will be in touch with a set of requests.

IMPORTANT: The editorial review process is now complete. PLOS will only permit corrections to spelling, formatting or significant scientific errors from this point onwards. Requests for major changes, or any which affect the scientific understanding of your work, will not be allowed.

Best regards,

Dr Omona Kizito, PhD

Academic Editor

PLOS Mental Health

Thank you for addressing the comments from our reviewers.